# Evaluation of Problem-Based Learning Experiences Addressing Health Promotion in the Fourth Grade of Primary School

**DOI:** 10.3390/children11070807

**Published:** 2024-07-01

**Authors:** Beatriz Mederer-Hengstl, Pilar Núñez-Delgado, Aurora Bueno-Cavanillas

**Affiliations:** 1Doctoral Programme in Education Sciences, University of Granada, 18012 Granada, Spain; bmederer@correo.ugr.es; 2Department of Didactics of Language and Literature, University of Granada, 18011 Granada, Spain; ndelgado@ugr.es; 3Department of Preventive Medicine and Public Health, University of Granada, 18016 Granada, Spain; 4CIBER of Epidemiology and Public Health (CIBERESP-Spain), 28029 Madrid, Spain; 5Instituto de Investigación Sanitaria de Granada (IBS-Granada), 18012 Granada, Spain

**Keywords:** health education, healthy habits, healthy diet, active learning strategies, problem-based learning, instrumental skills, attitudinal competences, primary school

## Abstract

Background: The global issues of overweight and sedentary lifestyles require comprehensive responses from health systems. Despite this, health education remains fragmented and sporadic. This study aims to target primary school children to instil healthy habits using active learning strategies, with a specific focus on promoting a healthy diet and physical activity. Methods: This study employed problem-based learning (PBL) projects with fourth-grade primary students to encourage the acquisition of healthy habits. Conducted over four consecutive years at a two-line school, the study utilised an opportunity sample, evaluating intervention and control groups by comparing their corresponding scores. The control groups consisted of 54 students, while the intervention group included 159 students. Data collection involved pre-test and post-test questionnaires to measure outcomes. Results: Differential evaluation scores revealed significantly higher results in the intervention groups compared to the control groups, with an average score increase of 3.62 points in control groups and 6.94 points in intervention groups, particularly in attitudinal competencies. The strategies employed facilitated the development of crucial skills such as information search, synthesis, representation, analysis, decision making, teamwork, and intrapersonal awareness. Additionally, they significantly broadened the knowledge acquired regarding healthy habits. Conclusions: The problem-based learning approach proved effective in helping children understand the importance of making healthy choices and encouraged the integration of such habits into their daily lives. These findings suggest that active learning strategies can significantly enhance health education and promote long-term healthy behaviours among primary school children.

## 1. Introduction

Childhood overweight and obesity are global health problems. This trend is growing everywhere. The prevalence of overweight (including obesity) among children and adolescents aged 5–19 has risen dramatically from just 8% in 1990 to 20% in 2022 [1,2]. Diet patterns in countries with a traditional healthy diet, such as China or Spain, are progressively farther from the original food, recipes, and cooking methods. Fast food is increasingly available, and demanded by children and young people [3]. At the same time, sedentary habits are increasing. Lack of physical activity and excessive time spent watching television or playing video games are the main drivers, together with dietary patterns, associated with the increasing trend of obesity and overweight [4,5,6,7]. In Spain, the Aladino study identifies 28.3% of 9-year-olds as sedentary, a percentage that reaches 33.4% among girls, with higher figures among children with obesity [8]. Public policies against these problems are insufficient, poorly funded, and even handicapped by food industry pressures [9].

Children’s obesity prevention must be embedded into health and education systems [10]. School health education should be a priority because of its universal scope, children’s unique learning power, and the opportunity to build healthy habits instead of replacing inappropriate ones [11].

However, health education is usually approached only broadly from the natural sciences area and is focused primarily on theoretical knowledge; more practical content is introduced from physical education, traditionally oriented towards accomplishing cardiovascular exercises [12]. Occasionally, educational programs from the health area are applied as extraordinary and extracurricular activities. Results obtained on these approaches are variable. Though usually favourable in the short term, they fail to modify behaviours effectively [13,14]. Some ambitious protocols have also been published [15]. However, they have yet to provide any results.

New teaching methodologies are now being considered to integrate learning objectives addressing content and skills. A good number of authors show and describe the positive impact that this kind of educational proposal has on students, including project-based learning and problem-based learning (PBL). PBL is defined as a task-centred teaching and learning modality developed in small groups. It is a shared process of negotiation between the participants, with the main objective being the achievement of a final product that promotes individual and autonomous learning within a work plan defined by objectives and procedures. Learners take responsibility for their own learning, discovering their preferences and strategies in the process [16,17]. The teacher assumes a role as a mediator and guide of their students, replacing the former teacher-instructor model [17,18].

PBL implementation is not limited to using already developed resources in the classroom but focuses on creating new ones [19]. On the contrary, it requires previous work, a careful and individualised dedication to each group, and consideration of individual needs. As far as we know, there are no reported experiences related to PBL applied to health, not only for achieving theoretical curriculum content but also for awakening children’s consciousness regarding their health. We aimed to evaluate the feasibility and impact of an innovative and active methodology in the teaching and learning processes, addressing the necessary curricular knowledge and awareness related to acquiring healthy habits.

## 2. Materials and Methods

We conducted an observational study in the fourth grade of primary school (9-year-old children). It was conducted at a charter school with two lines (25–27 students each) for four consecutive academic years. There was no follow-up after the children passed the fifth year. Although two different teaching strategies were compared, each teacher’s teaching plan was always respected, so no intervention was performed.

The chosen topic was related to knowledge about the human body, health, and hygiene, emphasising the importance of developing healthy habits. A meta-comprehension project was designed for the first year, redesigned and oriented towards the problem-based learning approach for the next one, and implemented with some variations over the following two years to meet and respond to the characteristics and needs of each group. To verify the effectiveness of this methodology, one of the lines was established as a control group, while the experience was implemented in the other one. In control groups, traditional teacher-centred methods were used.

### 2.1. Design of Learning Methods

PBL usually starts from the pedagogical model based on the Theory of Multiple Intelligences [20]. It depends on cooperative learning techniques and uses thinking-based learning strategies such as thinking routines, graphic organizers, and mind maps. Figure 1 displays the design of the questions generated and understanding goals.

The final objective of both PBL projects was to elaborate a weekly menu according to the needs of the students. Table 1 shows the performance activities established to acquire the necessary knowledge. These activities were classified into start-up, follow-up, assessment, and evaluation activities. Graphic organizers were provided to support students in capturing, searching, synthesising, and representing information. Mind mapping helped them to check, review, and self-assess the content to be learned. Students were also given several rubrics that allowed them to consider and assess all issues related to completing the different graphic organizers, individually and as a group. The rubrics and their achievement standards served as a guide to judge the quality of their work, both for teachers and for students. The use of rubrics throughout the entire teaching and learning process allowed the students to progress in achieving the levels.

### 2.2. Sample and Data Collection

An opportunity sample was used through four academic years, including all children in the fourth grade of primary school each academic year, from 2015–2016 to 2018–2019. The teacher who conducted the study taught both lines for two school years (the second and third of the four years encompassing this study) and only one line for the other two. Children in the second line, in the first and fourth years, formed the control group, which integrated 54 students. The exposed group included 159 students.

To evaluate the meta-comprehension project, we used the written test results established to control content acquisition (which is mandatory for progress assessment) and the improvement achieved when responding to the healthy habits questionnaire as a pre-test and post-test. This questionnaire was designed “ad hoc”. The control group followed the traditional teaching method based on the textbook and performed the same tests and questionnaires.

### 2.3. Data Analysis

We estimated the mean and 95% confidence intervals for the written test results. The pre- and post-test results were compared using the paired means comparison test. We applied two-way variance analysis to compare the differences obtained between groups, considering both year and intervention. Statistical analyses were performed using STATA software (version 15.0, StataCorp., LP, College Station, TX, USA).

### 2.4. Ethical Issues

We performed an observational study. The teaching methodology was registered in the annual scholar plan and approved by the school management. Only standard evaluation tools were used; to analyse and compare results, every child was identified by a number. No personal data were used.

## 3. Results

Table 2 summarises the mean pre- and post-tests, stratified by academic year and group. In all cases, the post-test results were significantly higher than the pre-test results, with a difference of 3.62 points (95% CI 3.23–4.00) in the control group and an average difference of 6.94 points (95% CI 6.32–7.56) in the exposed group (*p* < 0.001). While all groups improved their numbers of correct answers, this improvement was higher in the intervention groups (*p* < 0.001). This highlights that the baseline average scores obtained in the first year (2015–2016 academic year) were clearly lower than those obtained in subsequent years. However, the increase was still significantly higher in the exposed lines.

We analysed the answers to specific questions pre- and post-test. Table 3 shows the percentage of correct answers to pre- and post-test questions, stratified by group. The exposed group significantly improved on every question except 1, 6, and 11, all of which had a high score in the basal assessment. In the control group, the improvement was statistically significant for 11 of 23 questions. For questions 1, 3, 9, and 11, the basal score was already over 90%.

The acquisition of curricular content was not particularly modified according to the method developed. Table 4 shows the results (exam marks) achieved by each control and exposed group and the ranges of variation. They showed no statistical differences. However, the broader range in the control groups, where some students achieved lower scores, is worth noting.

## 4. Discussion

Our results show that by applying new methodologies, the acquisition of curricular content is similar to that in the teacher-centred groups. However, the pre–post comparison of the answers to the questionnaire showed a significantly better acquisition of health-related notions in the exposed group. For the questions addressing conceptual items, the percentages of correct answers improved in both the intervention and control groups. The exposed group improved in terms of the questions about attitudes or behavioural intention.

Strengths and limitations: We have designed and evaluated a problem-based learning approach focused on health: “Looking after yourself”. The guidelines applied and the final product are fully reproducible and can be adapted to different scholar levels. The mediating action of the teacher in the classroom must be considered. The teacher should have enough knowledge about the needs of the students, as well as a strong enough motivation regarding health, to offer them a better response and guidance, helping students to achieve more significant progress and commitment, especially for those students who need it most.

Our study has some limitations. We used an opportunity sample in just one primary school. Despite including multiple intelligences and skills in the project, our intervention did not modify the scholar environment, nor did it include parental involvement. Although we are aware of the importance of these factors, which have a large impact on health behaviour, this also means that this PBL approach could be applied in other sociocultural contexts with the same or similar effect. The observational nature of this study precludes the interchangeability of intervention and control groups. They were managed by different teachers, which also explains why we did not have a control group for two academic years. At the same time, it is possible that the results are biased by another teacher’s plan being different from the PBL experiences.

Another limitation is the lack of some evaluation in the long term. Nevertheless, the long-term effect depends on successive interventions throughout scholarly time and on parental involvement. All the studies reviewed show a weakness regarding the outcome measurement. Curriculum evaluation is focused on the cognitive area, so the knowledge acquired is easy to measure, but attitudinal achievements are subjectively evaluated. Health behaviour practices are sometimes assessed by self-administering questionnaires without any previous validation. We also measure attitudes through an “ad hoc” questionnaire. To permit the design, implementation, and evaluation of good teaching practices related to a healthy diet and physical activity, reaching a consensus about the relevant outcome and the method used to measure it is essential.

School-based programs represent an ideal setting to enhance healthy eating. Nevertheless, elementary school teachers often display low nutritional knowledge, low self-efficacy, and few skills for delivering nutrition education effectively [21]. There are too few structured scholarly interventions to promote healthy habits. Most of them are based on curricular content, frequently under the guidance of nutrition specialists or other healthcare professionals [22]; however, those that apply cross-curricular strategies or practical learning are more effective [21,23,24,25]. Other published studies show community interventions improve healthy behaviours. Talks delivered by health professionals add knowledge, but their effects are time-dependent [26,27]. Practical learning by applying cooking sessions, school gardening, or even taste [28] has been shown to increase children’s motivation related to their intake of fruits and vegetables [29,30]. Our study shows that, although all students improved their results, the increase was more significant in those who worked on projects. PBL helps with the understanding, organisation, and synthesis of information, apart from promoting other interpersonal skills, but it does not replace the time spent on individual work to consolidate new knowledge [31]. The questions in the questionnaires encompass curricular content and health knowledge as products of reflection and critical thinking. At this point, intervention groups evidenced significantly greater progress, supporting that PBL helps people learn to think and develop critical, analytical, and reflective thinking. Therefore, the strategies to be carried out must be explicitly instructed and guided, requiring effort and capacity from both students and teachers [31].

In an interesting review, Peralta et al. [21,23] show that resources developed for elementary school teachers to facilitate teaching healthy eating are scarce. Moreover, only some of them embed cross-curricular strategies, experimental learning strategies, or contingent reinforcement activities. The proposed PBL can be considered a cross-curricular strategy because it is developed in a transversal way into several curriculum subjects, and it uses experimental learning strategies (e.g., a research project in which children actively look for the ingredients of their usual food). PBL methodologies are feasible and can be developed with the usual available resources. This way, better and more homogeneous academic results can be achieved, and at the same time, students can be involved in critical consideration of related health issues. The final product integrates knowledge, beliefs, values, and attitudes, which are, in fact, the starting points for building healthy behaviour. We agree with Murimi et al. [24] regarding the necessity of multicomponent interventions that are age-appropriate and have an adequate duration. Nevertheless, each effort counts, and all teachers can increase the health awareness of their pupils.

## 5. Conclusions

We can conclude that healthy lifestyle promotion can be integrated into school curricula using problem-based learning. This methodology allows the acquisition of content and helps children understand the importance of healthy choices regarding food and physical activity and incorporate them into their daily lives. Every primary teacher can adapt the proposed model to their students’ level without additional resources.

## Figures and Tables

**Figure 1 children-11-00807-f001:**
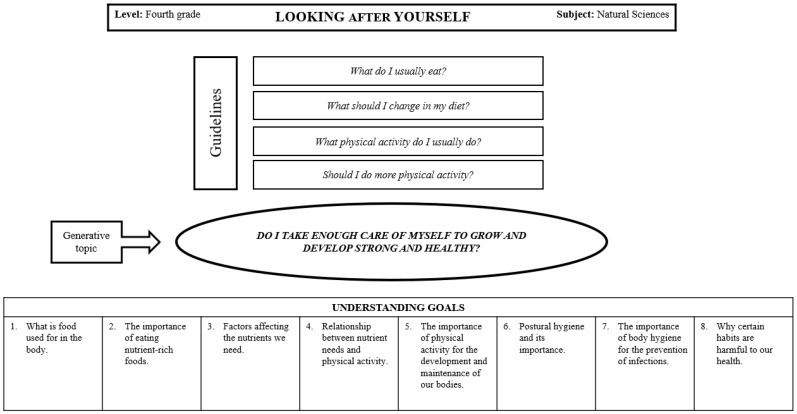
Global problem-based learning design.

**Table 1 children-11-00807-t001:** Looking after yourself: performance activities.

Summary	Grouping Cooperative Learning Techniques	Intelligences in Action
Start-Up Activities		
1st Session Physical Activity and Nutrition Log Evaluation: Spanish Questionnaire	Individual	Verbal–Linguistic Logical–Abstract Naturalistic Interpersonal–Intrapersonal
2nd Session Content: Healthy lifestyle habits. Brainstorming Why it is important to stay healthy? Graphic Organizer	Class Group	Verbal–Linguistic Intrapersonal Logical–Abstract Interpersonal–Intrapersonal
3rd Session Content: Food groups and nutritional needs. Thinking Routine: I See—I Think—I Wonder Food Pyramid Pooling Graphic Organizer	In Groups Round Robin	Verbal–Linguistic Logical–Abstract Visual–Spatial Interpersonal
Follow-Up Activities		
4th–5th Sessions Content: Nutritional components: carbohydrates, proteins, fats, vitamins, minerals, and water. Thinking Routine: I See—I Think—I Investigate Textos and Texts Cornwell Method	In Groups The Jigsaw	Verbal–Linguistic Logical–Abstract Visual–Spatial Interpersonal–Intrapersonal
6th Session Content: Functions of nutrients in the body. Thinking Routine: Cause—Effect—Conclusion What happens in our bodies?	In Groups	Verbal–Linguistic Logical–Abstract Visual–Spatial Interpersonal Intrapersonal
7th Session Content: Healthy and unhealthy food. Thinking Routine: Traffic light Which food is healthy? Debate Dr Seuss https://www.youtube.com/watch?v=B32u4qrIVZQ (Accessed on 8 February 2024)	In Groups Pens in the Middle Class Group	Logical–Abstract Visual–Spatial Intrapersonal Interpersonal Intrapersonal
8th Session Content: Consequences due to the lack of nutrients in our diet. Reading: Golosina y Perezoso by Willis Graphic Organizer Pooling Thinking Routine: Part-Part-Whole What happens if I do not eat…? Debate	In Groups Shared Reading Class Group	Verbal–Linguistic Logical–Abstract Interpersonal Intrapersonal
9th Session Content: Importance of a balanced and healthy diet, physical activity, and rest; alcohol and tobacco consumption. Reading: https://www.cuentosparaconversar.net/ http://magicfrogtales.com/5-free-short-stories-eat-healthy-be-healthy-and-smile/ (accessed on 8 February 2024)	Class Group	Verbal–Linguistic Interpersonal Intrapersonal
10th Session Content: Nutritional needs depending on age and physical activity. Thinking Routine: The Bridge Getting older: What are our teacher’s nutritional needs? Debate	In Groups 1, 2, 4 Class Group	Verbal–Linguistic Logical–Abstract Interpersonal Intrapersonal
11th Session Content: Importance of physical activity. Reading: ¿Quién menea el esqueleto? [Who’s shaking the skeleton?] Graphic Organizer Pooling and Debate	In Groups Shared Reading Class Group	Verbal–Linguistic Logical–Abstract Interpersonal Intrapersonal
12th Session Content: All discussed throughout the different sessions. Mind Map	Individual	Verbal–Linguistic Logical–Abstract Visual–Spatial Interpersonal Intrapersonal
13–14th Sessions Content: Healthy menus. Ideal Weekly Menu	In Groups	Verbal–Linguistic Logical–Abstract Naturalistic Interpersonal
15th Session Written and Oral Test Evaluation: English Questionnaire	Individual	Verbal–Linguistic Intrapersonal
Assessment Activities		
Personal Diary Graphic Organizer	Individual	Intrapersonal Verbal–Linguistic
Shared Reading	In Groups	Verbal–Linguistic Interpersonal
Individual Reading	Individual	Intrapersonal Verbal–Linguistic

**Table 2 children-11-00807-t002:** Results of pre-tests and post-tests stratified by group and academic year.

	Pre-Test Results Mean (CI 95%)	Post-Test Results Mean (CI 95%)	*p* Value	Mean Differences (CI 95%)
Academic year 2015–2016; Control group; N = 26	14.96 (13.93–16.00)	18.62 (17.45–19.78)	*p* < 0.0001	3.65 (3.18–4.13)
Academic year 2015–2016; Exposed group; N = 25	15.52 (13.95–17.09)	26.32 (25.84–26.80)	*p* < 0.0001	10.80 (9.38–12.22) ^1^
Academic year 2016–2017; Exposed group; N = 54	20.36 (19.61–21.11)	26.24 (25.84–26.64)	*p* < 0.0001	5.88 (5.06–6.70)
Academic year 2017–2018; Exposed group; N = 53	20.38 (19.67–21.10)	26.98 (26.56–27.40)	*p* < 0.0001	6.60 (5.96–7.23)
Academic year 2018–2019; Control group; N = 26	15.12 (13.85–16.38)	18.69 (17.32–20.06)	*p* < 0.0001	3.58 (2.98–4.18)
Academic year 2018–2019; Exposed group; N = 27	18.96 (17.77–20.15)	24.89 (23.47–26.31)	*p* < 0.0001	5.93 (3.91–7.94) ^2^
Control group; N = 52	15.04 (14.22–15.86)	18.65 (17.75–19.56)	*p* < 0.0001	3.62 (3.23–4.00)
Exposed group; N = 159	19.23 (18.65–19.80)	26.05 (25.58–26.52)	*p* < 0.0001	6.94 (6.32–7.56) ^3^

^1^ Mean differences were greater for exposed than for control group (*p* < 0.001). ^2^ Mean differences were greater for exposed than for control group (*p* = 0.0176). ^3^ Mean differences were greater for exposed than for control group (*p* < 0.001).

**Table 3 children-11-00807-t003:** Percentage of correct answers for any question asked pre- and post-test.

	Control Group (N = 52)			Exposed Group (N = 159)		
	Pre-Test	Post-Test	Statistical Signification	Pre-Test	Post-Test	Statistical Signification
1. Exercise must be done at least 2 or 3 times a week.	96.2%	98.1%	NS	86.2%	85.5%	NS
2. If we peel an apple it’s not necessary to wash it.	48.1%	48.1%	NS	52.8%	86.8%	*p* = 0.0000
3. Eating fresh fruit is the same as eating fruit in syrup or jam.	92.3%	94.2%	NS	79.9%	93.1%	*p* = 0.0001
4. An orange provides us with the same as its juice.	32.7%	32.7%	NS	27.0%	86.8%	*p* = 0.0000
5. Eating fruit by “biting” it, helps us to learn how to chew well.	63.5%	65.4%	NS	59.1%	88.7%	*p* = 0.0000
6. To grow strong and healthy you need to eat all kinds of food.	76.9%	84.6%	*p* = 0.0222	93.7%	91.8%	NS
7. A piece of fruit a day is enough to grow well.	50.0%	48.1%	NS	55.3%	86.8%	*p* = 0.0000
8. Walking to school is not doing exercise.	59.6%	61.5%	NS	79.2%	90.6%	*p* = 0.0012
9. It’s better not to get used to eating sweet desserts daily.	90.4%	94.2%	*p* = 0.0797	85.5%	93.1%	*p* = 0.0050
10. All juices are made from fresh fruit.	32.7%	34.6%	NS	45.9%	79.2%	*p* = 0.0000
11. Having fruit juice is the same as having soft fizzy drinks.	96.2%	100.0%	*p* = 0.0797	90.6%	92.5%	NS
12. It’s important to always have three courses at lunchtime.	53.8%	53.8%	NS	50.9%	79.2%	*p* = 0.0000
13. If we do not have enough calories, we will grow less and have less energy.	32.7%	50.0%	*p* = 0.0010	60.4%	88.1%	*p* = 0.0000
14. Proteins help us to grow healthy.	69.2%	98.1%	*p* = 0.0000	87.4%	93.1%	*p* = 0.0246
15. The mid-afternoon snack is not an important meal.	36.5%	36.5%	NS	54.7%	85.5%	*p* = 0.0000
16. A lack of vitamins or mineral salts causes illnesses.	63.5%	73.1%	*p* = 0.0119	72.3%	86.2%	*p* = 0.0001
17. Pasta has got a lot of vitamins and mineral salts.	17.3%	80.8%	*p* = 0.0000	57.2%	81.1%	*p* = 0.0000
18. Beans and pulses are rich in proteins.	23.1%	75.0%	*p* = 0.0000	71.7%	90.6%	*p* = 0.0000
19. Fruit and vegetables are the most important foods.	76.9%	94.2%	*p* = 0.0010	73.6%	86.2%	*p* = 0.0007
20. We can have large amounts of bread, because it’s very healthy.	44.2%	59.6%	*p* = 0.0018	60.4%	84.9%	*p* = 0.0000
21. We shouldn’t eat sweets and crisps.	51.9%	65.4%	*p* = 0.0034	68.6%	91.2%	*p* = 0.0000
22. Yogurt and cheese contain calcium.	67.3%	84.6%	*p* = 0.0010	88.1%	95.0%	*p* = 0.0036
23. Eggs don’t contain proteins.	25.0%	73.1%	*p* = 0.0000	74.2%	91.2%	*p* = 0.0000

**Table 4 children-11-00807-t004:** Average marks on the written test.

Academic Year	Control Group	Exposed Group
2015–2016	7.2 (3.3–9.6)	7.5 (4.7–10)
2016–2017	-	7.8 (4.4–9.9)
2017–2018	-	7.2 (5.0–9.9)
2018–2019	7.5 (3.7–10)	7.5 (4.7–10)

Two-way variance analysis comparing results by year and group was non-significant.

## Data Availability

The data presented in this study are available from the corresponding author upon request. The data are not publicly available due to privacy or ethical restrictions.

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
