# Peer review of "Evaluation of Problem-Based Learning Experiences Addressing Health Promotion in the Fourth Grade of Primary School"

_children, 2024, doi:10.3390/children11070807_

Round 1

Reviewer 1 Report

Comments and Suggestions for Authors

The manuscript was very interesting and valuable, but it needs basic correction, especially in the method and findings section.

Provide statistics on the lack of physical activity of children in the world and in the country.

Review the type of study. When you have made a change in teaching in a group, your study becomes experimental or semi-experimental, not observational.

How did you compare 159 people with 54 people? There is no correlation between the control group and the test group

How was the sampling method?

How was the sample size obtained?

  What software did you use to analyze the data?

  In table 2 and 3, add the amount of P value

Author Response

Thank you for your time, please, find enclosed the detailed answer to your queries

Reviewer 2 Report

Comments and Suggestions for Authors

Dear authors, first of all, congratulations for the proposal made. Steps must be taken to improve the health of children and young people.

Regarding the article, I have found some areas for improvement:

Abstract: it is usually advisable to indicate the objective of the paper. On the other hand, I read too much about PBL and not enough about health.

Introduction: I think that the target variables of studies are vaguely contextualized. The concept of health focuses exclusively on overweight, obesity and sedentary lifestyles. The concept of health in a broad sense is based on physical, psychosocial and emotional health. On the other hand, when health education is indicated, it would refer to healthy lifestyle habits (nutrition, physical activity, rest/sleep, sexuality and consumption of "tobacco and alcohol" substances). On the other hand, in the third paragraph when indicating the areas of knowledge in which the concept of health is worked, curricular proposals that in an interdisciplinary manner work on health are ignored, for example (Gerber et al. (2020); Santos-Beneit et al. (2019); Lopez et al. (2021)). When the PBL is contextualized, no citation is indicated to provide rigor to the information developed.

Materials and Methods: when it is indicated that the ownership of the center is "semi-private" I believe that the term to be used is "charter school". It is not clear whether the longitudinality of the study is with the same group or with different groups, always in 4th grade of primary school.

In the second paragraph of 2.1, a double final objective is indicated that is not in accordance with the objective formulated in the introduction. Are they secondary objectives? Is there an inconsistency?

Regarding figure and table 1, it is didactically correct but I am missing didactic elements that would provide relevant information. Objectives, competencies, contents, evaluation criteria. 

Ethical Issues: were parents consulted and asked for permission? Were the students in the control group "compensated"?

Results: the results are conditioned by the lack of a control group for two of the four years. The table of test results that is indicated as supplementary material I would add and interpret in the results. 

Discussion: it seems to me a mistake to begin the discussion with the strengths and limitations of the study. I still think it is sparse.

Conclusions: They are sparse and too obvious. Could it be a limitation that simply the methodology for novelty's sake generates greater interest in the experimental group?

Author Response

Thank you for your time. Please find enclosed the detailed answer to your queries.

Reviewer 3 Report

Comments and Suggestions for Authors

This study evaluates problem-based learning experiences addressed to health promotion. Although it handles with relevant issues, this manuscript also presents a bulk of aspects that must be improved:

1.   1. In general terms, this manuscript must be developed and better reasoned both in terms of theoretical framework and results achieved.

      2. Throughout the article, authors mention “two lines”. Can you please explain what do you mean by that?

    3.  Introduction must be developed. For example, a better definition of PBL must be provided as well as its potentials (there are lots of studies regarding that thematic).

4.     4.  Materials and methods sections must also be developed. What authors mean by “chosen topic” (line 66)?

5.      5.  Also, it is important to present a definition of PBL and the authors in which this work is based.

6.      6.  Table 1 must also be clearer.

7.     7.   How were instruments developed and validated? A better description of instruments must be provided.

8.      8.  Can you please explain what do you mean by observational study?

9.      9.  What statistical test was used to obtain the p-value?

     10. Results are presented in a confusing way. First there are some considerations regarding pre and post-test, then some concerning the written test and then again concerning pre and post-test.

     11. From the pre and post-test provided it is hard to infer conclusions regarding attitudes students’ improvement.

Author Response

(The authors gave the same response as above.)

Round 2

Reviewer 1 Report

Comments and Suggestions for Authors

Reply to comments is acceptable

And the manuscript is printable

thank you 

Reviewer 2 Report

Comments and Suggestions for Authors

Dear authors.

Thank you very much for taking into account the suggestions made, the paper has been sufficiently improved.